# Phenotypic effects of the U-genome variation in nascent synthetic hexaploids derived from interspecific crosses between durum wheat and its diploid relative *Aegilops umbellulata*

**Moeko Okada**[1], **Asami Michikawa**[1], **Kentaro Yoshida**[1], **Kiyotaka Nagaki**[2], **Tatsuya M. Ikeda**[3], **Shigeo Takumi**[1]*

**1** Graduate School of Agricultural Science, Kobe University, Kobe, Japan, **2** Institute of Plant Science and Resources, Okayama University, Okayama, Japan, **3** Western Region Agricultural Research Center, National Agriculture and Food Research Organization, Fukuyama, Hiroshima, Japan

* takumi@kobe-u.ac.jp

**Data Availability Statement:** The data sets supporting the results of this article are included

## Abstract

*Aegilops umbellulata* is a wild diploid wheat species with the UU genome that is an important genetic resource for wheat breeding. To exploit new synthetic allohexaploid lines available as bridges for wheat breeding, a total of 26 synthetic hexaploid lines were generated through crossing between the durum wheat cultivar Langdon and 26 accessions of *Ae. umbellulata*. In nascent synthetic hexaploids with the AABBUU genome, the presence of the set of seven U-genome chromosomes was confirmed with U-genome chromosome-specific markers developed based on RNA-seq-derived data from *Ae. umbellulata*. The AABBUU synthetic hexaploids showed large variations in flowering- and morphology-related traits, and these large variations transmitted well from the parental *Ae. umbellulata* accessions. However, the variation ranges in most traits examined were reduced under the AABBUU hexaploid background compared with under the diploid parents. The AABBUU and AABBDD synthetic hexaploids were clearly discriminated by several morphological traits, and an increase of plant height and in the number of spikes and a decrease of spike length were commonly observed in the AABBUU synthetics. Thus, interspecific differences in several morphological traits between *Ae. umbellulata* and *A. tauschii* largely affected the basic plant architecture of the synthetic hexaploids. In conclusion, the AABBUU synthetic hexaploid lines produced in the present study are useful resources for the introgression of desirable genes from *Ae. umbellulata* to common wheat.

## Introduction

To mitigate the effects of climate change and increasing human population, which are increasingly important problems facing humanity, the use of natural variations in wild species is required for crop breeding [1–3]. Wild wheat relatives, including *Aegilops* species, carry abundant natural variations and have been used for wheat breeding through synthetic hexaploids

within the article. Files containing raw sequence data for the RNA sequencing are available in the sequence read archive of DDBJ (accession numbers DRA007097 and DRA006404).

**Funding:** This work was supported by Grants-in-Aid for Scientific Research (B) No. 16H04862 and for Scientific Research on Innovative Areas No. 19H04863 from the Ministry of Education, Culture, Sports, Science, and Technology of Japan, and by grant 2018-7 from the Iijima Foundation to ST.

**Competing interests:** The authors have declared that no competing interests exist.

and alien chromosome introgression lines [4]. Common wheat (*Triticum aestivum* L., AABBDD genome) is an allohexaploid species derived from a natural crossing between tetraploid wheat (*Triticum turgidum* L., AABB) and *Aegilops tauschii* Coss. (DD) [5,6]. Artificial crosses of tetraploid wheat and *Ae. tauschii* can reproduce synthetic hexaploid wheat with the AABBDD genome. The D-genome donor species *Ae. tauschii* has a large distribution, from Turkey to China, and most accessions of *Ae. tauschii* belong into either of two major lineages, TauL1 or TauL2 [7,8]. The *Ae. tauschii* population carries many intraspecific variations in traits including heading time, spike and grain morphology, ABA sensitivity, and hybrid incompatibility with tetraploid wheat [9–13]. These variations in *Ae. tauschii* are useful for wheat breeding through introgression from synthetic wheat hexaploids with the AABBDD genome [4,14,15].

To overcome problems caused by severe environmental stress, other *Aegilops* species could be used for breeding new wheat cultivars. *Aegilops umbellulata* Zhuk., a diploid wild relative species with the U genome, has been utilized for wheat breeding as a genetic resource of disease resistance genes and grain quality-related genes [16–19]. Recently, genotype-by-sequencing techniques have facilitated the genetic analysis of disease resistance and linkage map construction in *Ae. umbellulata*, although no reference genome sequence data is available [20,21]. RNA-seq is also a powerful tool for detecting single nucleotide polymorphisms (SNPs) and developing novel molecular markers, not only in wheat relatives with reference genome sequences, but also in strains without any reference genome information [22–28]. Although *Ae. umbellulata* has a narrow distribution from Greece to Iraq, its genetic diversity and number of alleles with rare frequencies are higher than in *Ae. tauschii* [26]. Pollen of *Ae. umbellulata* can be crossed to tetraploid wheat. In about 50% of cross combinations between the tetraploid wheat cultivar Langdon (Ldn) and various *Ae. umbellulata* accessions, the $F_1$ hybrids with the ABU genome show one of two types of hybrid growth abnormalities, severe growth abortion (SGA) or grass-clump dwarfism (GCD) [29]. Synthetic wheat hexaploids with the AABBUU genome can be obtained from ABU $F_1$ hybrids showing normal growth and be used as bridges to introduce useful traits from *Ae. umbellulata* into common wheat, including disease resistance genes *Lr76* and *Yr70* [30]. Synthetic hexaploids with the AABBUU genome generally generate hard grains, suggesting that *Ae. umbellulata* variations in grain quality-related traits are useful for the enlargement of grain hardness diversity in hard-textured common wheat [31].

Allopolyploidization is frequently accompanied by genetic and epigenetic modifications in the synthetic allopolyploid genomes of wheat and *Arabidopsis* [32–36]. Moreover, phenotypic traits of the synthetic allopolyploids are affected by epistatic interactions among their subgenomes. The D-genome variations of some phenotypic traits observed at the diploid level are not necessarily expressed in synthetic hexaploid wheat lines with the AABBDD genome, and the variations of other traits are narrower in the synthetic lines than in the parental *Ae. tauschii* accessions [37]. Similarly, greater epistatic alteration of gene expression levels occurs in allopolyploid wheat compared with their parental accessions [34,38]. Thus, genetic diversity in *Ae. umbellulata* should be evaluated under the allohexaploid background of the AABBUU synthetic lines. Here, to evaluate the U-genome variations under the allohexaploid background, we independently produced 26 synthetic lines derived through interspecific crossing between Ldn and 26 *Ae. umbellulata* accessions, and then agricultural traits were measured in the synthetic hexaploid lines and *Ae. umbellulata* accessions. Based on the results, we also discuss the distinct effects on the examined traits in the synthetic allohexaploids between the U and D genomes.

## Materials and methods

### Plant materials

In total, 26 *Ae. umbellulata* accessions from seeds supplied by the National BioResource Project-Wheat, Japan (https://shigen.nig.ac.jp/wheat/komugi/) were propagated from a single plant by self-pollination (Table 1). A tetraploid wheat accession *T. turgidum* ssp. *durum* cv. Langdon (Ldn) was used as the female parent and crossed with each of the 26 accessions of *Ae. umbellulata* (Fig 1). All 26 synthetic hexaploid wheat lines with the AABBUU genome (ABU hexaploids, $F_2$ generation) were generated by 0.1% colchicine (Wako Pure Chemical Industries, Osaka, Japan) treatment for 5 h at the seedling stage in each $F_1$ triploid hybrid ($F_1$ generation). Thus, the synthetics share the A and B genomes from Ldn and contain the U genome derived from diverse *Ae. umbellulata* accessions. All synthetics grew normally in a greenhouse at Kobe University (34˚43'N, 135˚13'E), and none showed hybrid growth abnormalities such as SGA and GCD [29]. Four lines of synthetic hexaploid wheat with the AABBDD genome (ABD hexaploids), Ldn/KU-2097 (Syn6214), Ldn/IG126387 (Syn6240), Ldn/PI476874 (Syn6256), and Ldn/KU-2069 (Syn6262), were also used in this study. These four ABD hexaploids showed various heading/flowering time, and did not exhibit any growth abnormalities [37,39]. The ABD hexaploids were grown under the same conditions as the ABU hexaploids.

### Chromosome preparation and genomic *in situ* hybridization

After imbibition of mature seeds of synthetic hexaploids in tap water overnight, they were incubated overnight at 4˚C. Seeds were germinated at 23˚C. Root tips were incubated for 22 h at 4˚C to synchronize cell division, then fixed in acetic acid:ethanol (1:3) at room temperature for 2 days. After fixation, root tips were stained by acetocarmine and squashed in 45% acetic acid under a cover slip. The cover slips were removed on dry-ice, and the glass slides were dried.

Genomic *in situ* hybridization (GISH) analysis of mitotic metaphase chromosomes was performed using a fluorescence *in situ* hybridization protocol as previously described with minor modifications [40]. Genomic DNA was extracted from young leaves of the *Ae. umbellulata* accession KU-4074. A DNA probe was labeled by nick translation using a DIG-Nick Translation Mix (Roche Diagnostics, Basel, Switzerland) and digested by *Hae*III at 37˚C for 5 h. After hybridization with the chromosome samples, the DIG-labeled probe was visualized using a rhodamine-conjugated anti-digoxigenin antibody (Roche Diagnostics). Chromosomes were counterstained with 0.1 µg/ml 4,6-diamino-2-phenylindole (DAPI). GISH signals and DAPI stained chromosomes were captured using a fluorescence microscope (Axioskop2, Carl Zeiss, Oberkochen, Germany) coupled with a chilled charge-coupled device camera (Axiocam HR, Carl Zeiss), and images were pseudo-colored and processed using ZEN software blue edition (Carl Zeiss).

**Table 1. List of *Ae. umbellulata* accessions used to produce synthetic hexaploid lines.**

| Origins | Accession No. |
|---|---|
| Azerbaijan | KU-2932 |
| Greece | KU-12186, KU-12198 |
| Iran | KU-4109 |
| Iraq | KU-4001, KU-4006, KU-4007, KU-4010, KU-4017, KU-4024, KU-4026, KU-4030, KU-4035, KU-4039, KU-4043, KU-4046, KU-4068 |
| Turkey | KU-4070, KU-4074, KU-4075, KU-4080, KU-4081, KU-4087, KU-4103, KU-12200, KU-12204 |

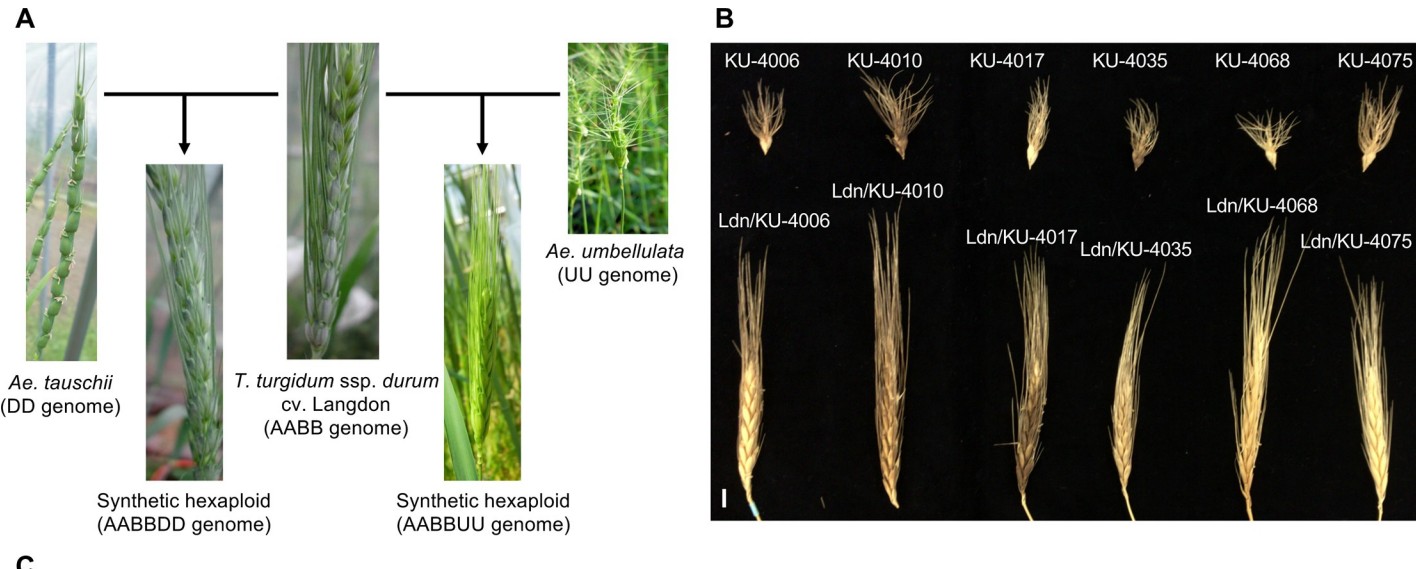

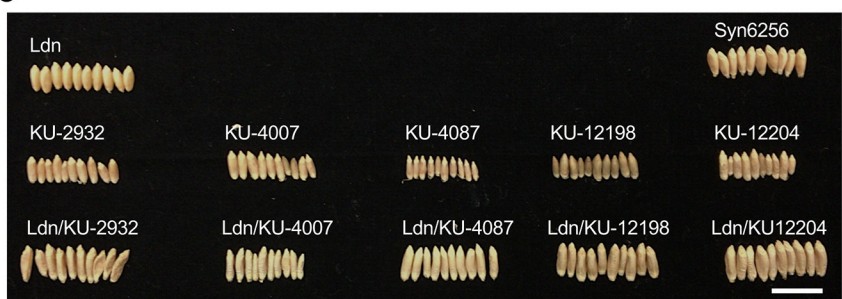

**Fig 1. Photos of Ldn, *Ae. umbellulata*, *Ae. tauschii*, ABU hexaploids, and ABD hexaploids.** (A) Spike morphology of the synthetic hexaploid lines and their parental lines. (B,C) Comparison of spike morphology (B) and seed shape (C) between the ABU hexaploids and their parental *Ae. umbellulata* accessions. Scale bar corresponds to 1 cm.

## U chromosome-specific marker development and genotyping

To develop molecular markers for specifically detecting the U-genome chromosomes in the AABBUU genome, we used RNA sequence (RNA-seq) reads of Ldn and 12 accessions of *Ae. umbellulata* that were obtained in our previous report [26,28] (accession numbers DRA007097 and DRA006404). These RNA-seq reads were aligned to the *de novo* assembled transcripts of *Ae. umbellulata* KU-4017, which were anchored to the reference genome sequences of *Ae. tauschii* [26]. Assuming genome collinearity between *Ae. tauschii* and *Ae. umbellulata*, chromosomal positions of *Ae. umbellulata* transcripts were estimated based on their chromosome positions of *Ae. tauschii* [26]. SNP calling was conducted according to our previous report [26] by estimating nucleotide substitutions between the AB genomes of Ldn and the U genome. Since the nucleotide substitution sites were monomorphic in the 12 *Ae. umbellulata* accessions, they allowed us to discriminate between the U and AB genomes. Based on these nucleotide substitutions, PCR markers and cleaved amplified polymorphism sequence (CAPS) markers were designed (S1 Table). The chromosomal positions of the nucleotide substitutions and the designed markers were visualized on the *Ae. tauschii* genome using R software ver. 3.6.1 (https://www.R-project.org/). Total DNA was extracted from the leaves of Ldn, the parental *Ae. umbellulata* accessions, and synthetic hexaploid lines. For genotyping, 40 cycles of PCR were performed using the Quick Taq HS DyeMix (TOYOBO, Osaka, Japan) and

the following conditions: 10 s at 94˚C, 30 s at appropriate annealing temperature (58˚C or 60˚C), and 45 s at 68˚C. After digestion with restriction enzymes as shown in S1 Table, the PCR products and their digests were resolved in 2% agarose gels and visualized under UV light after staining with ethidium bromide.

## Phenotype measurement and statistical analysis

Phenotypic traits were measured in seasons 2016–2017 and 2017–2018. In total, 39 traits as listed in S2 Table were measured using four plants for each synthetic line. Seeds of the *Ae. umbellulata* accessions and synthetics (F$_3$ generation) were sown in November of each year, and the two plants were grown in each pot arranged randomly. All morphological traits of the synthetic hexaploid lines and the parental *Ae. umbellulata* accessions were measured in the three earliest tillers of each plant. Abbreviations of the examined traits are listed in S2 Table. The first and second florets of the lowest, central, and top spikelets were evaluated to measure the spikelet- and awn-related traits. Heading and flowering dates (HD and FD) were recorded as days after sowing.

The seed-related traits, Grain area size (AS), Perimeter length of grain (PL), Grain length (GL), Grain width (GW), Length-width-ratio of grain (GLWR) and Circularity (CS), were measured using *SmartGrain* software ver. 1.2 [41], and mean values were calculated using data from four plants of each synthetic line. The grain hardness of the ABU hexaploids and Ldn were measured by a single kernel characterization system (SKCS 4100, Perten, Stockholm, Sweden). The grain hardnesses of the four ABD hexaploids were referred to from our previous study [31]. These data were statistically analyzed using R Studio ver. 1.2.1335 software (http://www.rstudio.com) with R software ver. 3.6.1. Student's *t* test was used to compare the ABU and ABD hexaploids. Statistical differences were assessed with the two-side test with an alpha level of 0.05. The correlations among the morphological traits that were estimated based on Pearson's correlation coefficient values and principal component (PC) analyses were conducted using R software ver. 3.6.1.

## Results

### Chromosome numbers of the synthetic ABU hexaploids

In total, 26 synthetic lines were generated through interspecific crossings between Ldn and 26 accessions of *Ae. umbellulata* (Fig 1). These synthetic lines produced self-pollinated seeds. Chromosome elimination sometimes results in severe phenotypic abnormalities in synthetic allopolyploids [42]. To evaluate the somatic chromosomes of the ABU synthetic hexaploid lines, GISH analysis was conducted (Fig 2). In the GISH results, 42 somatic chromosomes were observed in the root cells of the F$_3$ plants as expected. GISH analysis using *Ae. umbellulata* DNA as probes efficiently distinguished chromosomes of the U genome from those of the AB genome. Fourteen chromosomes of the U genome were visualized in the 42 chromosomes of the ABU polyploids.

To confirm whether all synthetic hexaploids contained the U-genome chromosomes, molecular markers were designed based on previous RNA-seq data [26,28]. In total, 16,481 nucleotide substitutions were detected between the U and AB genomes, and were distributed throughout all the chromosomes (Table 2, Fig 3). We designed chromosome-specific PCR and CAPS markers that discriminated between AB genomes and the U genome. To determine whether all ABU hexaploids contained a set of the U-genome chromosomes, one marker per chromosome was constructed (Fig 3). All 26 synthetic ABU hexaploids in which the phenotypic traits were measured were genotyped with these markers. We confirmed that all tested synthetic hexaploids contained a set of the U-genome chromosomes (Fig 4).

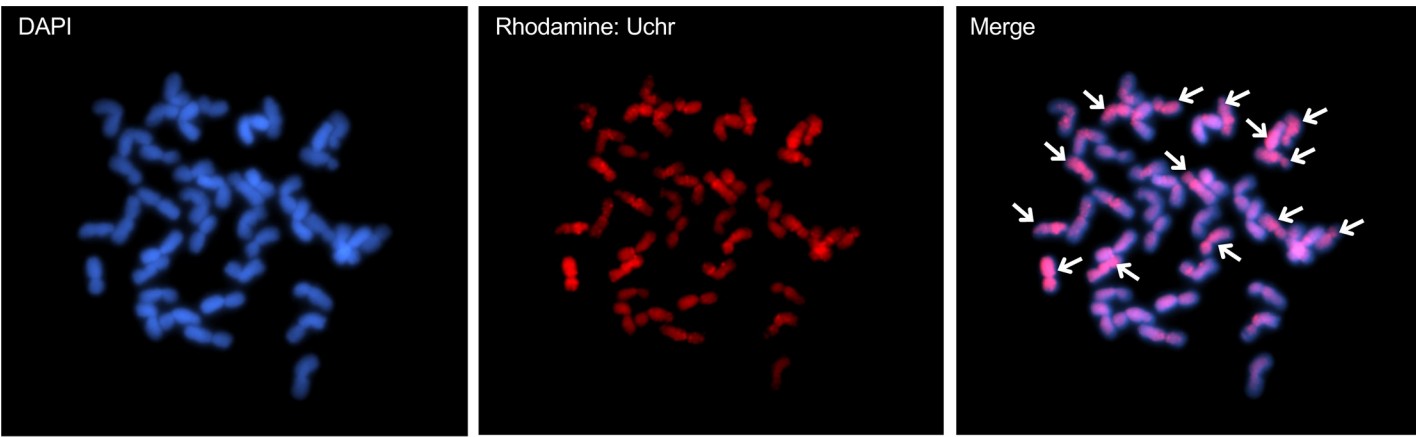

**Fig 2. Genomic *in situ* hybridization analysis of synthetic hexaploids with the AABBUU genome.** Chromosomes were hybridized by *Ae. umbellulata* GISH probe (red) and stained by DAPI (blue). Arrows indicate 14 U-genome chromosomes out of 42 chromosomes.

## Phenotypic variations of the ABU hexaploids and their parental *Ae. umbellulata* accessions

To phenotypically characterize the newly synthesized allohexaploid lines with the AABBUU genome, four synthetic hexaploid plants (the $F_3$ generation) were grown for each line in seasons 2016–2017 and 2017–2018. Spikes breaking off as a unit occurred in all of the 26 ABU hexaploids at the grain maturation stage, although the AB genome donor Ldn showed a non-brittle rachis phenotype. Thus, the shattering pattern of the ABU hexaploids was transmitted from *Ae. umbellulata* in which the spikes breaking off as a unit was commonly observed.

Two flowering and 37 morphology-related traits were measured in Ldn, the ABU hexaploids, parental *Ae. umbellulata* accessions, and the ABD hexaploids. The *Ae. umbellulata* accessions and the ABU hexaploids showed large variations in flowering and morphology-related traits (Figs 1 and 5, S1 Fig, S2 Table). The ranges in heading time (HD) and flowering time (FD) were respectively 20 and 15 days among the ABU hexaploids. These ranges were similar to those previously observed for ABD hexaploids [37]. The large variations in the parental *Ae. umbellulata* accessions were well maintained in the ABU hexaploids, whereas the

**Table 2. Summary of the number of SNPs used for molecular marker development.**

| Accession | # of SNPs against KU-4017 transcripts | # of non-redundant SNPs | # of fixed SNPs between Ldn and *Ae. umbellulata* |
|---|---|---|---|
| Ldn | 521584 | 40944 | 16481 |
| KU–4017 | 1461 | | |
| KU–4026 | 3317 | | |
| KU–4035 | 17584 | | |
| KU–4043 | 12038 | | |
| KU–4052 | 13003 | | |
| KU–4103 | 5795 | | |
| KU–5934 | 25866 | | |
| KU–5954 | 14990 | | |
| KU–12180 | 30564 | | |
| KU–12198 | 24172 | | |
| KU–8–5 | 14326 | | |
| KU–8–7 | 25564 | | |

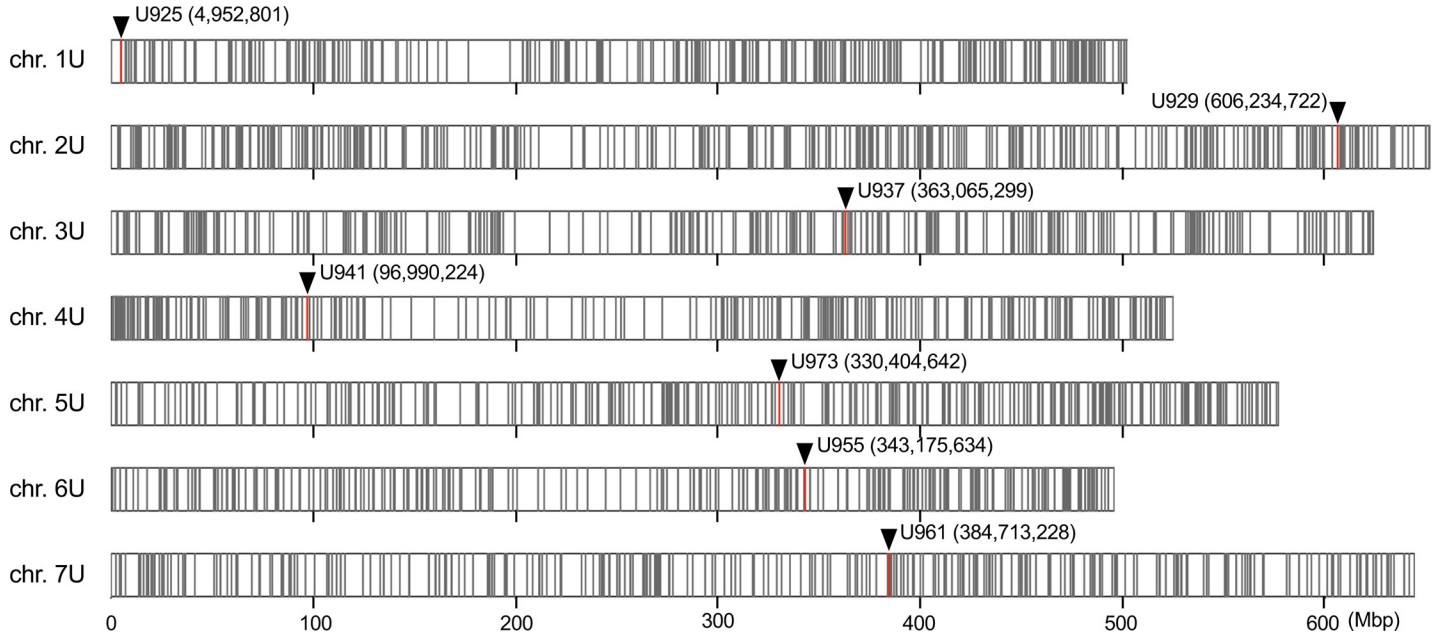

**Fig 3. Chromosomal distribution of nucleotide substitutions between the U and AB genomes on the physical map of *Ae. tauschii* (scale; Mb).** Positions of the nucleotide substitutions were estimated based on chromosomal positions of *Ae. tauschii* where *Ae. umbellulata* transcripts were anchored. Red bars and arrowheads show the chromosomal positions of molecular markers developed in the present study.

ranges of variation in most of the measured traits were reduced under the AABBUU hexaploid background (S2 Table). Significant correlations were observed in flowering, spike, and grain-related traits between two growth seasons, whereas no significant correlation between the seasons was detected in plant height-related traits (S2 Fig).

Bottom awn length showed the highest correlation coefficient value between the ABU hexaploids and their parental *Ae. umbellulata* accessions in the 39 examined traits (S2 Table). Significant correlations between the ABU hexaploids and their parental *Ae. umbellulata* accessions were observed in 13 traits that were related to flowering time, stem width, spikelet shape, awn length, and grain morphology. Although these trait variations were well expressed under the AABBUU hexaploid background, the ranges of variations in the ABU hexaploids were less than those observed in *Ae. umbellulata* (S2 Table). The spike morphology was clearly distinct among Ldn, ABU hexaploids, and ABD hexaploids (Fig 1), and large variations were observed in spike-related traits among the parental *Ae. umbellulata* accessions and were well maintained in the ABU hexaploids (Fig 5, S1 Fig). However, no significant correlation was observed in spike-related traits between diploid and hexaploid backgrounds (Fig 6, S3 Fig).

Next, to examine the effects of the U and D genomes on flowering time and morphology under the hexaploid background, the flowering and morphology-related traits were compared between the ABU and ABD hexaploids. Most of the traits showed significant differences between the ABU and ABD hexaploids, whereas no significant difference was observed in FD, HD, InL4, SpW1, SpW3, AL1, GW, and CS (Fig 7, S4 Fig). In particular, spike length, the number of spikelets, and the number of spikes were remarkably distinct between the ABU and ABD hexaploids. These results indicated that variations of flowering and grain-related traits were mainly affected by the genetic effects of the AABB parent, and that variations in spike-related traits were largely influenced by the U and D genomes added to the AABB genome.

Correlation coefficients were compared among the traits examined (Fig 8). In *Ae. umbellulata*, high negative correlations were observed among flowering traits and leaf morphologies,

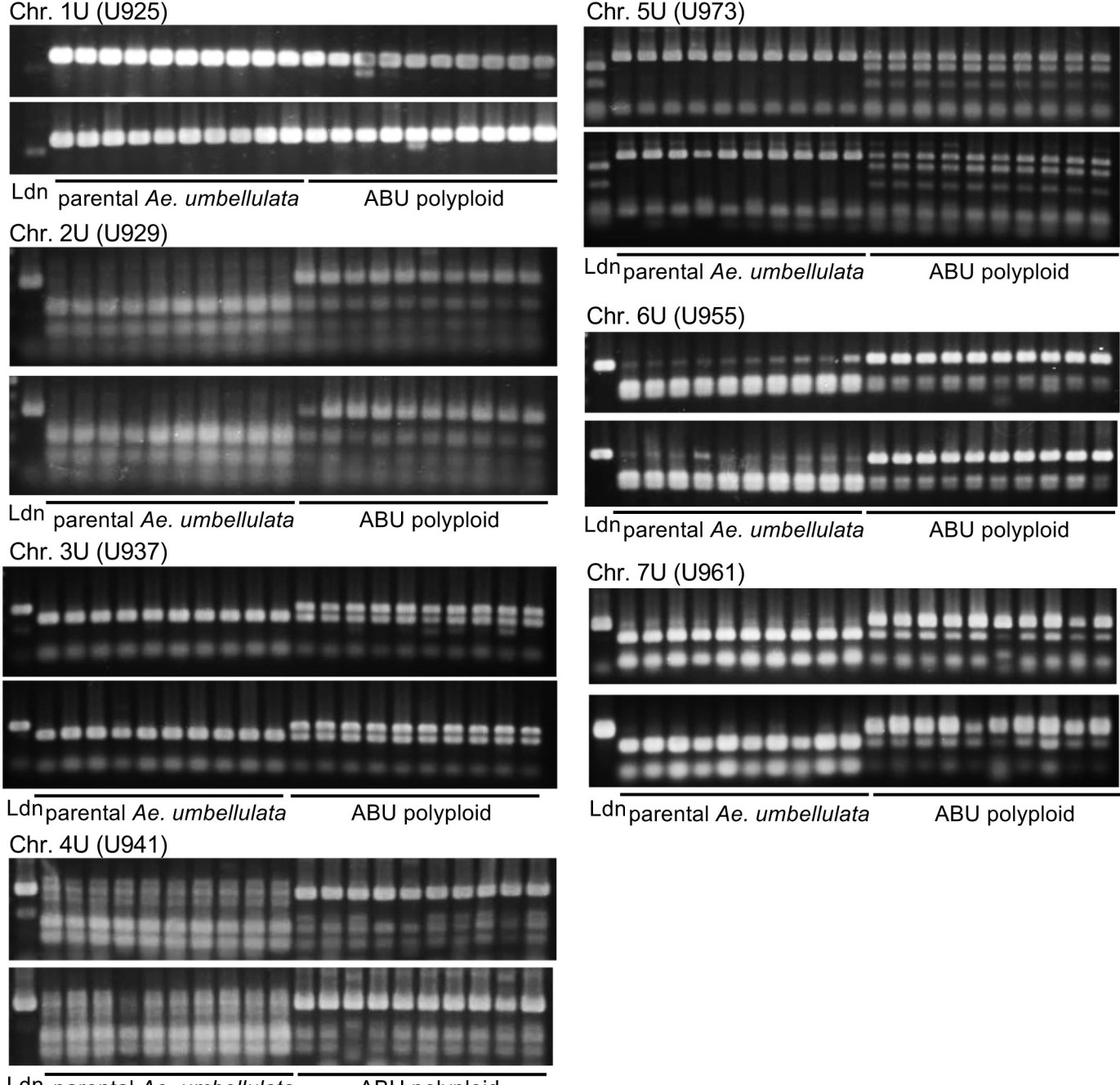

**Fig 4. Confirmation of the U genome chromosomes in the synthetic hexaploid lines using the developed PCR and CAPS markers.** Marker names are shown on the left of each gel image. The presence/absence of PCR products between the AB and U genomes was detected in the U925 marker (chromosome 1U). The size differences between the AB and U genomes were observed in the other markers.

and positive correlations were observed among leaf morphologies and grain shapes. These correlations were reduced or became the opposite in ABU hexaploids. However, higher correlations were observed among the awn lengths and grain shapes in ABU hexaploids compared to *Ae. umbellulata*. Thus, the relationships between traits were dramatically different between parental *Ae. umbellulata* accessions and ABU hexaploids.

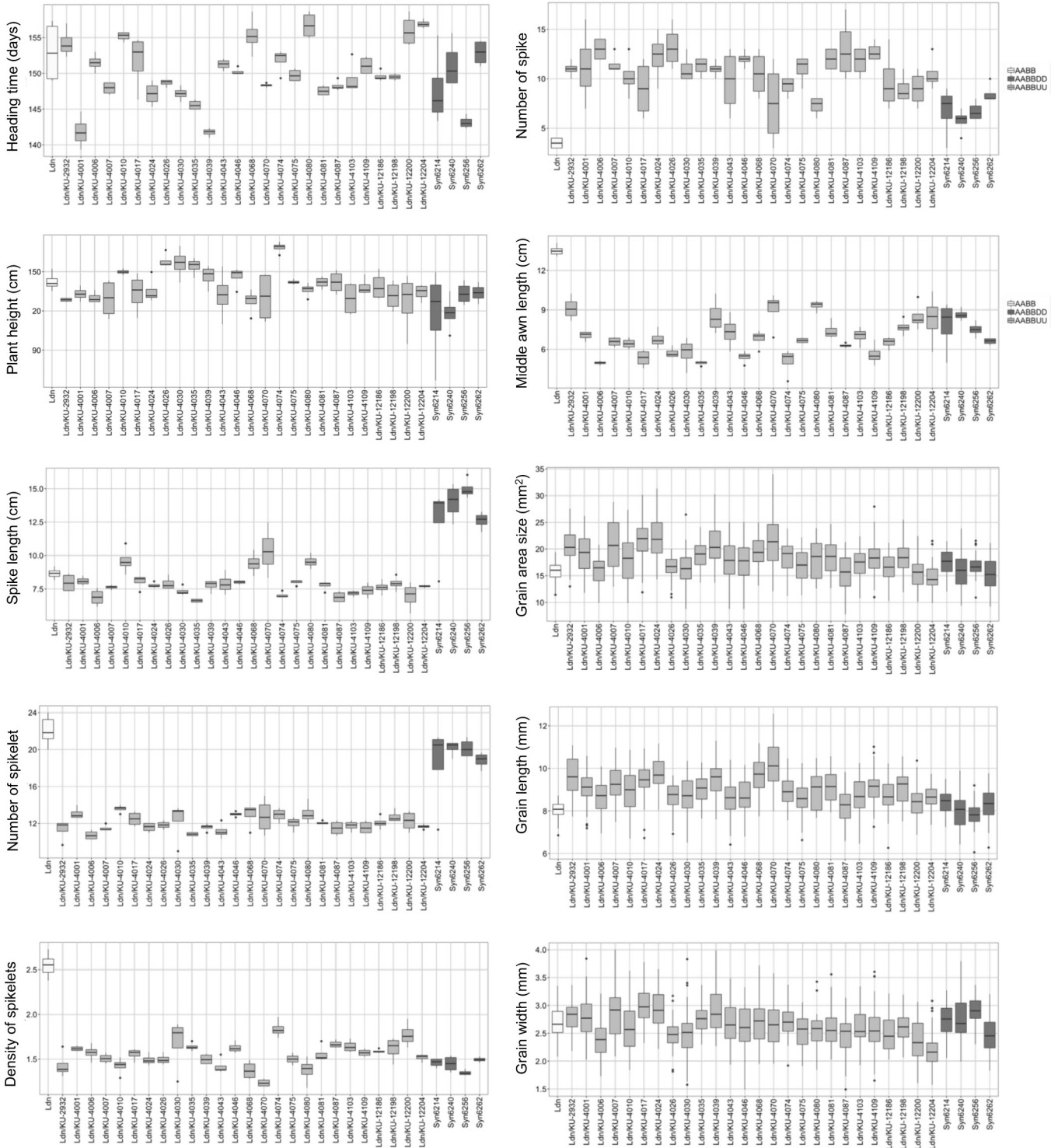

**Fig 5. Phenotypic comparisons of heading time (HD), plant height (PH), spike length (SL), number of spikelets (SpN), density of spikelets (SpD), number of spikes (SN), middle awn length (MAL), grain area size (AS), grain length (GL), and grain width (GW) in the synthetic hexaploid lines with the AABBUU and AABBDD genomes.**

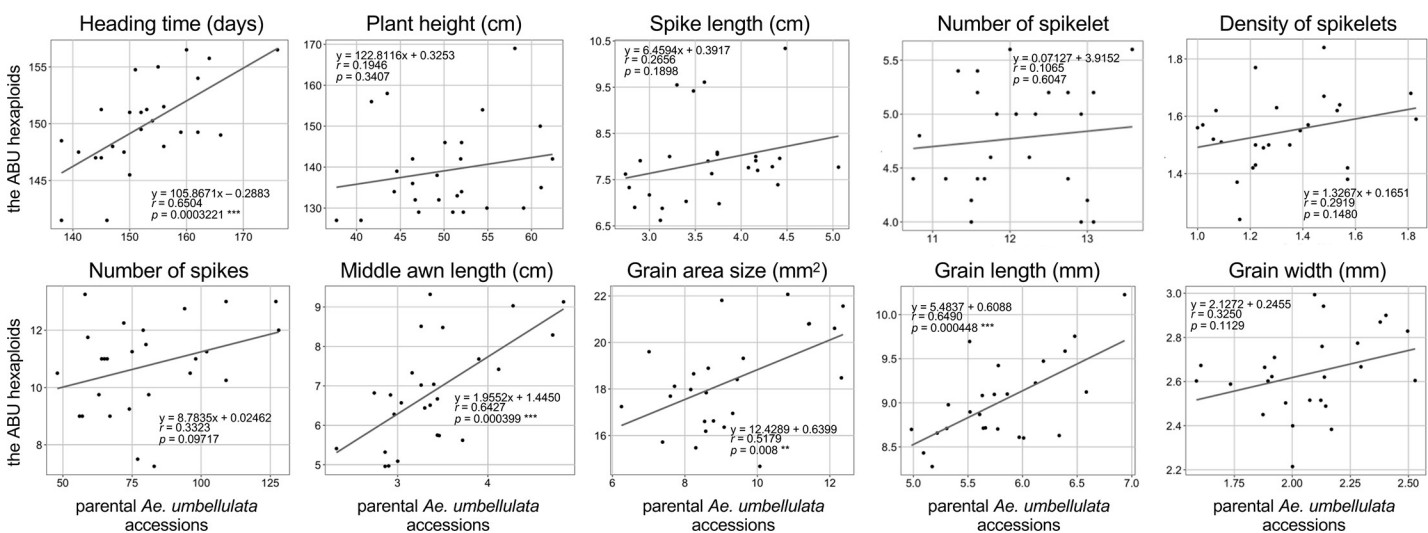

**Fig 6. Comparisons of variability and correlations in HD, PH, SL, SpN, SpD, SN, MAL, AS, GL, and GW between the ABU hexaploids and their parental *Ae. umbellulata* accessions.** The regression line and the correlation coefficient for each plot are indicated. Significant correlation coefficients are marked by asterisks (**$p < 0.01$, ***$p < 0.001$).

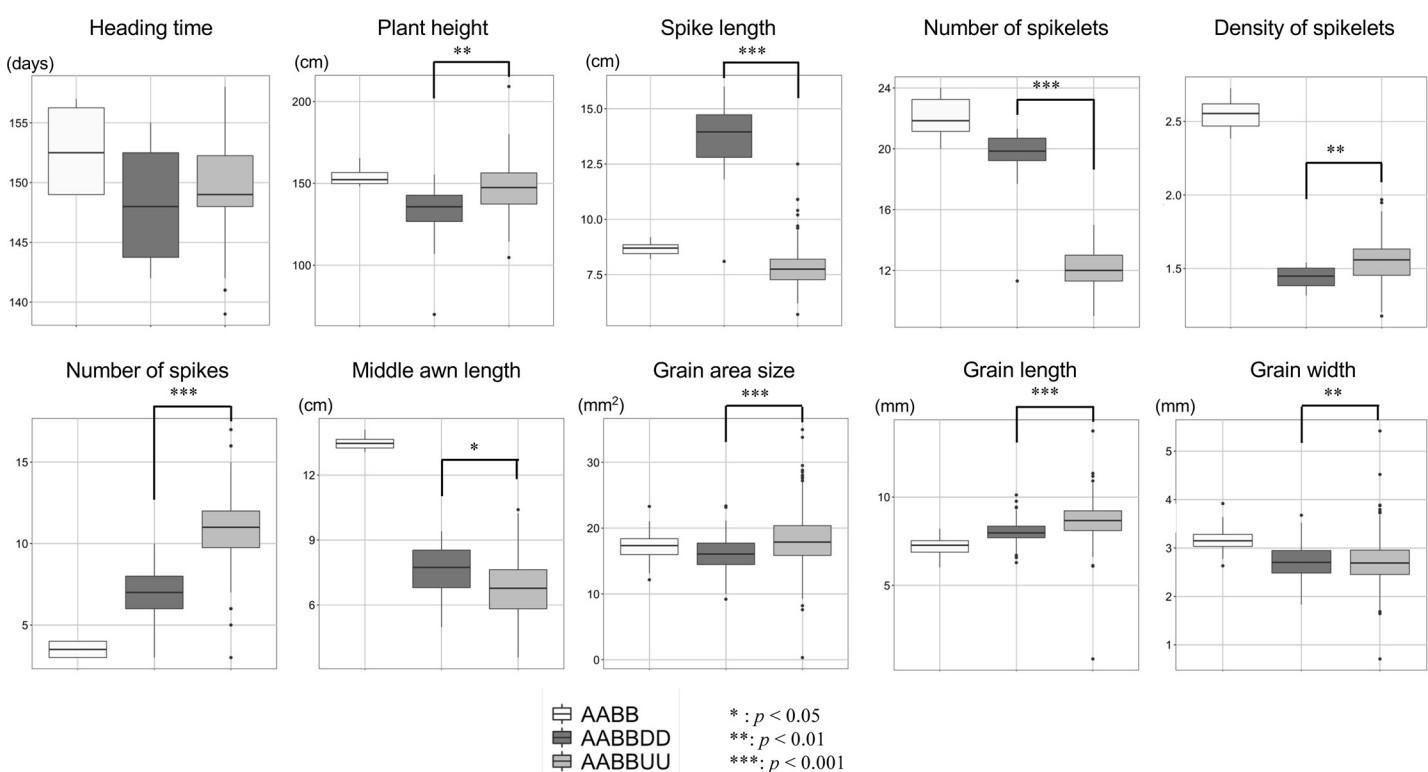

**Fig 7. Phenotypic comparisons between the ABU and ABD hexaploids in HD, PH, SL, SpN, SpD, SN, MAL, AS, GL, and GW.** Asterisks indicate significant differences (*$p < 0.05$, **$p < 0.01$, ***$p < 0.001$).

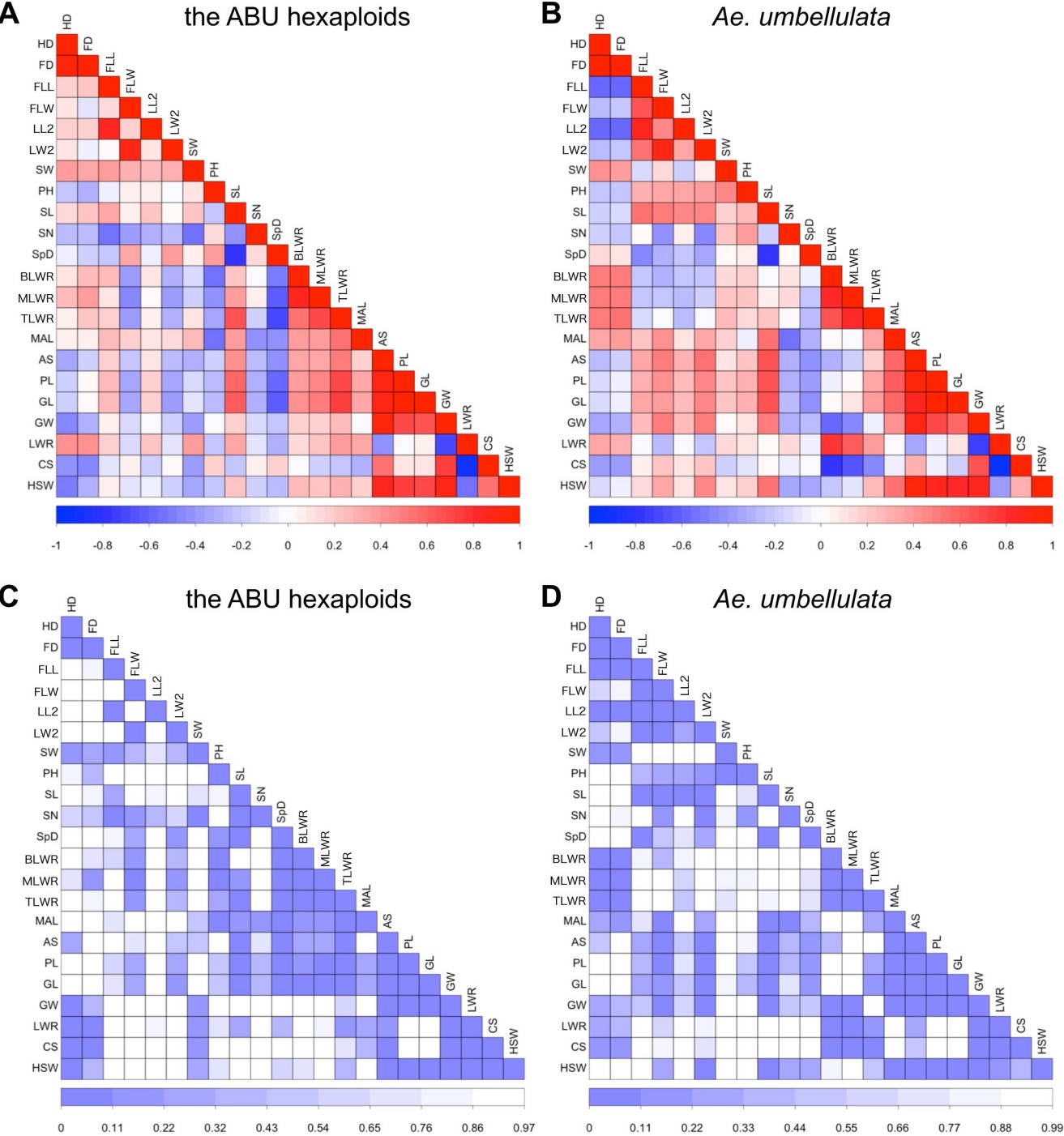

**Fig 8.** Correlations among the examined traits in the ABU hexaploids (A,C) and their parental *Ae. umbellulata* accessions (B,D) based on Pearson's coefficient values (A,B) and *p*-values (C,D). The darker colors indicate higher coefficients.

## Principal component (PC) analysis of the estimated traits

PC analysis was conducted using all trait data obtained from Ldn, the ABU hexaploids, and the ABD hexaploids. The contributions of the first two principle components, PC1 and PC2, were respectively 31.1% and 18.4% (Fig 9). The variation in PC1 values had a large effect on

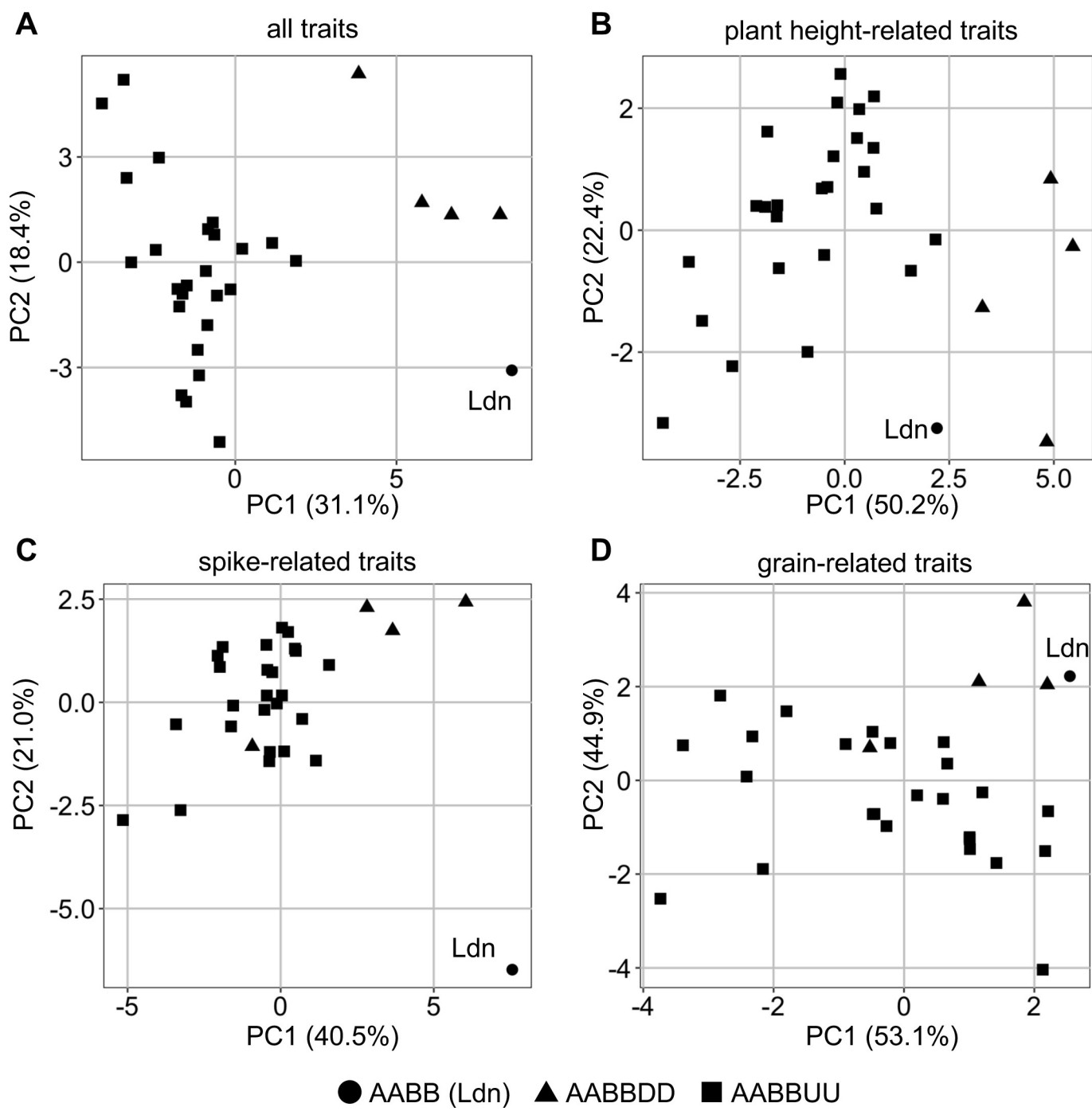

**Fig 9. Principal component analysis of morphological traits in synthetics with AABBUU genomes, synthetics with AABBDD genomes, and Ldn.** Scatter diagrams of the principal components were based on the first (PC1) and second (PC2) components. (A) All examined traits. (B) Plant height-related traits. (C) Spike-related traits. (D) Seed-related traits.

the leaf morphologies (FLL, FLW, LL2, and LW2) with positive eigenvectors and on SW, SN, and MAL with negative eigenvectors (S3 Table). The variations in PC2 and PC3 were influenced by the plant height-related traits (PH and InL1 to 5) and by the flowering time (HD and FD), respectively. A scatter plot based on the values of PC1 and PC2 in all examined traits

showed clear differentiation among Ldn, the ABU hexaploids, and the ABD hexaploids (Fig 9A). The ABU hexaploids were distinguished from others by their PC1 values, and the PC2 values discriminated between Ldn and the ABD hexaploids.

Next, PC analyses were conducted using data of plant height-related traits (FLL, FLW, LL2, LW2, SW, 1 to 5 InL, StL, and PH), spike-related traits (SL, SN, SpN, SpD, BSpL, BSpW, MSpL, MSpW, TSpL, TSpW, BLWR, MLWR, TLWR, BAL, MAL, and TAL), and grain-related traits (AS, PL, GL, GW, GLWR, and CS). A scatter plot of PC1 and PC2 values in plant height-related trait showed that PC1 values distinguished the ABU hexaploids from the ABD hexaploids and Ldn (Fig 9B). The PC1 and PC2 values based on the spike-related traits clearly discriminated between Ldn and the synthetic hexaploids (Fig 9C). A synthetic AABBDD hexaploid line, Syn6240, was not distinguished from ABU hexaploids by PCA of the spike-related traits. The length and width of spikelets in Syn6240 were closer to those of the ABU hexaploids than those of the other ABD hexaploids (S2 Table). PC1 and PC2 values based on the grain-related traits showed no difference among Ldn, the ABU hexaploids, and the ABD hexaploids (Fig 9D). A synthetic AABBDD hexaploid line, Syn6262, was not distinguished from the ABU hexaploids by PCA in grain-related traits. Grains were longer and the GRWL value was larger in Syn6262 than in the other ABD hexaploids and similar to those in the ABU hexaploids (S2 Table). These results indicated that the U and D genomes added to the AABB genome had large effects on spike- and grain-related traits under a hexaploid background.

## Grain hardness of synthetic hexaploids

Grain hardness is an important trait for determining wheat grain quality and is mainly controlled by two puroindoline (PIN) genes, *Pina* and *Pinb* [43]. Nucleotide variations in *Pin* genes are observed among *Ae. umbellulata* accessions, and some *Ae. umbellulata* accessions have hard-textured grains [31]. To evaluate the variability of grain hardness in the ABU hexaploids, mature seeds of Ldn and 19 lines of the ABU hexaploids in which sufficient amounts of grains were obtained were analyzed by SKCS and compared with the ABD hexaploids. The hardness data of five ABD hexaploids was referred to from our previous study [31]. Grains of the ABD hexaploids were soft, and their hardness indexes were lower than 50. On the other hand, the hardness indexes of the ABU hexaploids varied from 59.15 to 88.01 (Fig 10, S4 Table). Grains of Ldn/KU-4039 showed the highest value (88.01) among those of the ABU hexaploids, and the value of Ldn/KU-4039 was similar to that of Ldn (87.66).

## Discussion

In the present study, 26 synthetic hexaploid lines with the AABBUU genome were successfully generated through ABU triploids that were obtained by interspecific crossing of the durum cultivar Ldn with 26 accessions of *Ae. umbellulata*. The ABU hexaploids showed wide variations in flowering and morphology-related traits and exhibited no abnormal growth phenotypes, such as hybrid lethality or hybrid weakness, which are frequently observed in triploid hybrids between tetraploid wheat and *Ae. umbellulata* [29]. RNA-seq is a powerful tool to detect SNPs even in non-reference species such as *Ae. umbellulata* [26]. To confirm the somatic chromosome numbers in the ABU hexaploids, molecular markers were developed using the RNA-seq data of *Ae. umbellulata* [26] and the reference genome sequence of *Ae. tauschii* [44]. Nucleotide substitutions between the U and AB genomes were distributed throughout the seven chromosomes, and the U-genome chromosomes were clearly distinguished from the AB genome by the developed markers (Figs 3 and 4). If the RNA-seq-based markers can cover an entire chromosome, the chromosome segments transmitted from various *Aegilops* species into tetraploid and hexaploid wheat cultivars can be identified during the

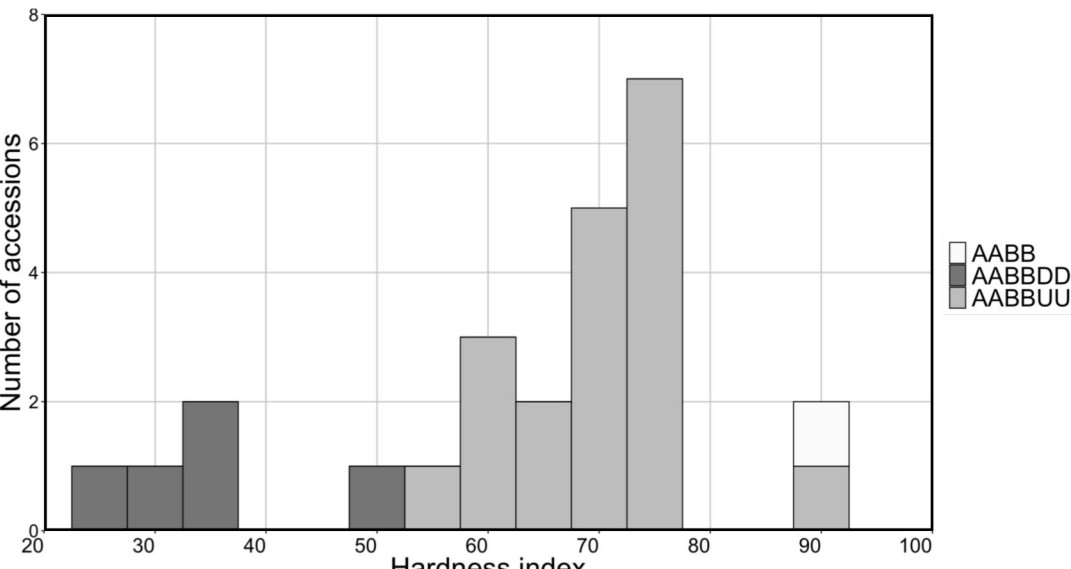

**Fig 10. Histogram of SKCS hardness index in 19 lines of synthetic hexaploids with AABBUU genomes, five lines of synthetic hexaploids with AABBDD genomes, and Ldn.** Nineteen lines of synthetic hexaploids with AABBUU genomes were selected by selfed-seed fertility. The hardness index data of the ABD hexaploids was referred to in our previous study [31].

wheat breeding process. Thus, the marker information will contribute to wheat breeding through introgression of alien chromosome segments from relatives.

Large variations in 39 traits were observed in both the ABU hexaploids and their parental *Ae. umbellulata* accessions (S2 Table, Fig 5). The variations observed in the ABU hexaploids generally originated from the U genome, because the A and B genomes of the ABU hexaploids were commonly derived from the cultivar Ldn. Therefore, the ABU hexaploids are useful not only to introduce desirable genes from *Ae. umbellulata* into wheat cultivars, but also to evaluate the possibility of transmitting target phenotypes and the expression of genetic variations in *Ae. umbellulata* under an allohexaploid background. According to our comparison of 39 morphological and flowering-related traits between the ABU hexaploids and their parental *Ae. umbellulata* accessions, positive correlations were significantly observed in flowering- and grain-related traits (Fig 6). Although the spike morphology-related traits in the ABU hexaploids were significantly different from Ldn and the ABD hexaploids (Fig 7), no significant correlation was observed in plant height- and spike-related traits (Fig 6). These correlation patterns between synthetic hexaploids and their diploid parents were somewhat different from observations in the AABBDD synthetic hexaploids and their *Ae. tauschii* accessions [37]. No correlation was commonly observed in internode lengths both between the ABU hexaploids and their parental diploid accessions and between the ABD hexaploids and their parental ones. For morphological traits with no correlation, the AB genome should have larger effects than the diploid U and D genomes. A few major genes on the AB genome would strongly control the morphological traits with no correlation and would mask the variations transmitted from the U and D genomes under the allohexaploid backgrounds. Sometimes strong major genes hide the effects of minor quantitative trait loci in regard to morphological phenotypes [45].

In addition, the variation ranges in the *Ae. umbellulata* accessions were decreased under the AABBUU hexaploid background compared with those under the parental diploid species (Fig 6). Repression of the variation ranges observed under the allohexaploid background was previously observed in *Ae. tauschii* [37]. The differences in expression patterns of the

phenotypic variations under the distinct polyploidy levels could be due to buffer effects of the shared AB genome and epistatic interactions between the AB genome and the added genome. Moreover, the degree of the buffer effects of the shared genome appears to depend on the traits. Thus, the ABU hexaploids do not necessarily reflect the natural variations of *Ae. umbellulata* in unchanged conditions, but the *Ae. umbellulata* variations could be useful to alter many morphological traits such as spike-, spikelet-, and grain-related traits in wheat breeding.

The ABU and ABD hexaploids were clearly discriminated by several morphological traits (Figs 7 and 9). In the ABU hexaploids, increases in plant height and in the number of spikes and a decrease of spike length were commonly observed (Fig 7), whereas no significant correlation was observed in these traits between the ABU hexaploids and their parental *Ae. umbellulata* accessions (Fig 6). In addition, grain hardness was also clearly distinct between the ABU and ABD hexaploids (Fig 10). The phenomenon of spikes breaking off as a unit was specific to the ABU hexaploids. The phenotypic differences between the ABU and ABD hexaploids indicate distinct effects of the U and D genomes on phenotypes of the synthetic hexaploids. Interspecific differences of phenotypes between *Ae. umbellulata* and *A. tauschii* should largely affect the basic plant architecture and grain hardness of the synthetic hexaploids. Transmission of genetic factors related to the interspecific differences between *Ae. umbellulata* and common wheat could greatly alter the plant architecture and grain quality.

To clarify the changes in gene expression patterns accompanied by the transmission of desirable phenotypes from *Ae. umbellulata* to the AABBUU synthetic hexaploid, further studies are required. Gene expression patterns are stochastically and epigenetically changed during the generation of allopolyploid *Arabidopsis* and wheat [34–36,38]. The altered gene expression patterns in allopolyploids include homoeolog expression bias, changes in alternative splicing patterns, and altered expression levels of small RNAs [46–48]. However, information on the altered gene expression patterns during the generation of synthetic wheat hexaploids is limited even in the ABD hexaploids, and there is little known about the ABU hexaploids. Therefore, the set of nascent ABU hexaploid lines produced in the present study represents a useful resource for understanding the altered gene expression patterns and genetic and epigenetic changes during the generation of synthetic hexaploids.

## Supporting information

**S1 Table. Primers used in this study.**
(DOC)

**S2 Table. Variations in morphological traits in Ldn, synthetics, and corresponding *Ae. umbellulata* parental accessions.**
(DOC)

**S3 Table. Eigenvectors for PC1, PC2, and PC3 among Ldn and the ABU and ABD hexaploids based on all morphological and spikelet- and grain-related traits examined.**
(DOC)

**S4 Table. Grain characters in synthetic hexaploids with the AABBUU genome and Ldn as measured by SKCS.** The SKCS data of the other ABU hexaploids were referred to our previous study [31].
(DOC)

**S1 Fig. Phenotypic comparisons of heading time (HD), plant height (PH), spike length (SL), number of spikelets (SpN), density of spikelets (SpD), number of spikes (SN), middle awn length (MAL), grain area size (AS), grain length (GL), and grain width (GW) in the**

**parental *Ae. umbellulata* accessions.**
(TIF)

**S2 Fig. Comparisons of variability in all examined traits between measured seasons.** FLL, FLW, LL, LW, and SW were measured only in season 2017–2018.
(TIF)

**S3 Fig. Comparisons of variability in all examined traits between the ABU hexaploids and their parental *Ae. umbellulata* accessions.**
(TIF)

**S4 Fig. Phenotypic comparison of all traits examined between the ABU and ABD hexaploids.**
(TIF)

## Acknowledgments

We thank Dr. Kanenori Takata at the Western Region Agricultural Research Center of NARO for his help in measuring grain hardness. We also thank Dr. Atsushi Torada at the HOKUREN Agricultural Research Institute for teaching us the method of colchicine treatment.

## Author Contributions

**Conceptualization:** Shigeo Takumi.

**Formal analysis:** Moeko Okada, Kentaro Yoshida.

**Funding acquisition:** Shigeo Takumi.

**Investigation:** Moeko Okada, Asami Michikawa, Kiyotaka Nagaki, Tatsuya M. Ikeda, Shigeo Takumi.

**Resources:** Shigeo Takumi.

**Supervision:** Shigeo Takumi.

**Writing – original draft:** Moeko Okada, Shigeo Takumi.

**Writing – review & editing:** Kentaro Yoshida, Kiyotaka Nagaki, Shigeo Takumi.

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
