## [Decision Letter · Decision Letter 0]

2 Mar 2020

PONE-D-20-03425

Phenotypic effects of the U-genome variation in nascent synthetic hexaploids derived from interspecific crosses between durum wheat and its diploid relative Aegilops umbellulata

PLOS ONE

Dear Dr. Takumi,

Thank you for submitting your manuscript to PLOS ONE. After careful consideration, we feel that it has merit but does not fully meet PLOS ONE’s publication criteria as it currently stands. Therefore, we invite you to submit a revised version of the manuscript that addresses the points raised during the review process.

We would appreciate receiving your revised manuscript by Apr 16 2020 11:59PM. To enhance the reproducibility of your results, we recommend that if applicable you deposit your laboratory protocols in protocols.io, where a protocol can be assigned its own identifier (DOI) such that it can be cited independently in the future. For instructions see: http://journals.plos.org/plosone/s/submission-guidelines#loc-laboratory-protocols

We look forward to receiving your revised manuscript.

Kind regards,

Ajay Kumar

Academic Editor

PLOS ONE

Journal Requirements:

Reviewers' comments:

Reviewer's Responses to Questions

**Comments to the Author**

1. Is the manuscript technically sound, and do the data support the conclusions?

Reviewer #1: Yes

2. Has the statistical analysis been performed appropriately and rigorously? 

Reviewer #1: Yes

3. Have the authors made all data underlying the findings in their manuscript fully available?

Reviewer #1: Yes

4. Is the manuscript presented in an intelligible fashion and written in standard English?

Reviewer #1: Yes

5. Review Comments to the Author

Reviewer #1: The manuscript by Okada et al. 2020 describes the phenotypic variation observed in newly generated hexaploid lines developed from tetraploid durum wheat and diploid Aegilops umbellulata, a wild relative of the ancestral wheat genomes. The authors used the durum line Langdon for the AABB genomes and 26 accessions from 5 different countries as sources for the UU genome. GISH and U-genome specific markers were used to confirm the presence of the U-genome chromosomes in each of the 26 newly developed hexaploids.

Overall the manuscript is well written with the exception of a few lines. Technically, the experiment design and analysis are sound as described. This study and the germplasm developed for it will likely be valuable for the wheat breeding and research community. Minor comments and concerns are listed below.

The resolution on several of the figures is low, this needs to be improved for them to be of value. In particular, Figures 1 , 5, 6, and 8 need work. Additionally, the symbols in the principal component plots (Figure 9) are difficult to see and need to be enlarged.

I believe it is PLOS one policy to have abbreviations defined when they first appear in text. Considering the large number of abbreviations in this study, perhaps the supplementary table with all abbreviations could be placed in the materials and methods.

p. 5, line 107: It is unclear to me what “representative” means here. Please elaborate on the criteria used to select the synthetic AABBDD hexaploid lines.

p. 7, line 164: The word “repeatedly” can be removed here since it is implied if the same phenotypic traits were measured in both growing seasons.

p. 8, lines 165-167: This would read better if you discussed the number of plants per line first, then the number of plants per pot.

p. 8, line 173: The “seed related” traits measured using SmartGrain software should be listed here.

p. 11, line 243: Can an explanation for no significant correlation for plant height between seasons be provided. In my experience this is an unusual result for wheat.

p. 16, line 362: “data-based” should probably be “data-base”.

p. 17, line 385: “mask” would be a clearer word than “cover”.

p.17, lines 386-387: The sentence “Sometimes strong major genes hide the effects of minor quantitative trait loci in regard to morphological phenotypes.” should be supported with citations or removed.

6. PLOS authors have the option to publish the peer review history of their article (what does this mean?). If published, this will include your full peer review and any attached files.

Reviewer #1: No

---

## [Author Response · Author response to Decision Letter 0]

3 Mar 2020

Dear Editors and reviewers,

Thank you very much for the reviewers’ kind comments and suggestions to our manuscript.

We examined all of the comments, and revised our manuscript according to the comments as followed.

Improved sentences and words were marked by red color in text.

In addition, we fit the format of our manuscript to the PLoS ONE’s style requirements. Our gel blot images are reported in a submission’s figures but not in Supporting Information files.

To Reviewer #1,

Thank you very much for your understanding our article and for giving us polite comments.

1. The resolution on several of the figures is low, this needs to be improved for them to be of value. In particular, Figures 1, 5, 6, and 8 need work. Additionally, the symbols in the principal component plots (Figure 9) are difficult to see and need to be enlarged.

<response> We confirmed the quality of submitted figures (original tif files). Their quality was high. In addition, we improved Fig. 9 with enlarged the plotted symbols. 

2. I believe it is PLOS one policy to have abbreviations defined when they first appear in text. Considering the large number of abbreviations in this study, perhaps the supplementary table with all abbreviations could be placed in the materials and methods.

<response> We included the abbreviation list for examined traits in S2 Table of Supporting information. In addition, a following sentence was inserted in the second section of M&M; ‘Abbreviations of the examined traits are listed in S2 Table’.

3. p. 5, line 107: It is unclear to me what “representative” means here. Please elaborate on the criteria used to select the synthetic AABBDD hexaploid lines.

<response> We omitted ‘representative’, and added one following sentence; ‘These four ABD hexaploids showed various heading/flowering time, and did not exhibit any growth abnormalities [37,39]’ to explain the characteristics of these four lines.

4. p. 7, line 164: The word “repeatedly” can be removed here since it is implied if the same phenotypic traits were measured in both growing seasons.

<response> We omitted ‘repeatedly’.

5. p. 8, lines 165-167: This would read better if you discussed the number of plants per line first, then the number of plants per pot.

<response> We exchanged the two sentences.

6. p. 8, line 173: The “seed related” traits measured using SmartGrain software should be listed here.

<response> We inserted the seed-related traits as followed; ‘The seed-related traits, Grain area size (AS), Perimeter length of grain (PL), Grain length (GL), Grain width (GW), Length-width-ratio of grain (GLWR) and Circularity (CS), were measured using SmartGrain software ver. 1.2 [41]’.

7. p. 11, line 243: Can an explanation for no significant correlation for plant height between seasons be provided. In my experience this is an unusual result for wheat.

<response> This question is difficult to answer. Little effect of the pollen parents was observed on the plant height under the hexaploid synthetic backgrounds. This means that plant height could be affected by the AB parental cultivar. Therefore, we think that the unusual result is due to the Langdon’s characteristics. Langdon is sensitive to the environmental growth conditions.

8. p. 16, line 362: “data-based” should probably be “data-base”.

<response> We changed this tem to ‘the RNA-seq-based’.

9. p. 17, line 385: “mask” would be a clearer word than “cover”.

<response> We changed from ‘cover’ to ‘mask’.

10. p.17, lines 386-387: The sentence “Sometimes strong major genes hide the effects of minor quantitative trait loci in regard to morphological phenotypes.” should be supported with citations or removed.

<response> We added a reference; Nguyen et al. (2015).

We believe that the revised manuscript is now suitable for publication. We look forward to hearing from you at your earliest convenience.

Yours sincerely,

Shigeo Takumi

(Corresponding author)

---

## [Decision Letter · Decision Letter 1]

18 Mar 2020

Phenotypic effects of the U-genome variation in nascent synthetic hexaploids derived from interspecific crosses between durum wheat and its diploid relative Aegilops umbellulata

PONE-D-20-03425R1

Dear Dr. Takumi,

We are pleased to inform you that your manuscript has been judged scientifically suitable for publication and will be formally accepted for publication once it complies with all outstanding technical requirements.

With kind regards,

Ajay Kumar

Academic Editor

PLOS ONE

Additional Editor Comments (optional):

Reviewers' comments:

Reviewer's Responses to Questions

**Comments to the Author**

1. If the authors have adequately addressed your comments raised in a previous round of review and you feel that this manuscript is now acceptable for publication, you may indicate that here to bypass the “Comments to the Author” section, enter your conflict of interest statement in the “Confidential to Editor” section, and submit your "Accept" recommendation.

Reviewer #1: All comments have been addressed

2. Is the manuscript technically sound, and do the data support the conclusions?

Reviewer #1: Yes

3. Has the statistical analysis been performed appropriately and rigorously? 

Reviewer #1: Yes

4. Have the authors made all data underlying the findings in their manuscript fully available?

Reviewer #1: Yes

5. Is the manuscript presented in an intelligible fashion and written in standard English?

Reviewer #1: Yes

6. Review Comments to the Author

Reviewer #1: I would like to thank the authors for addressing all the comments and concerns in the first review. I have no further edits to the manuscript, which in my opinion is well written and provides relevant information on a valuable resource for wheat researchers, pathologists, and breeders. The time and effort you put into this manuscript are appreciated, the review process has been a pleasant one.

7. PLOS authors have the option to publish the peer review history of their article (what does this mean?). If published, this will include your full peer review and any attached files.

Reviewer #1: No

---

## [Editor Report · Acceptance letter]

20 Mar 2020

PONE-D-20-03425R1 

Phenotypic effects of the U-genome variation in nascent synthetic hexaploids derived from interspecific crosses between durum wheat and its diploid relative *Aegilops umbellulata*

Dear Dr. Takumi:

I am pleased to inform you that your manuscript has been deemed suitable for publication in PLOS ONE. Congratulations! Your manuscript is now with our production department. 

With kind regards,

on behalf of

Dr. Ajay Kumar 

Academic Editor

PLOS ONE